# Ultrasound-Assisted Extraction of Carotenoids from Orange Peel Using Olive Oil and Its Encapsulation in Ca-Alginate Beads

**DOI:** 10.3390/biom11020225

**Published:** 2021-02-05

**Authors:** Ivana M. Savic Gajic, Ivan M. Savic, Dragoljub G. Gajic, Aleksandar Dosic

**Affiliations:** 1Faculty of Technology in Leskovac, University of Nis, Bulevar Oslobodjenja 124, 16000 Leskovac, Serbia; savicivan@tf.ni.ac.rs; 2Gajic Associates, Doktora Pantica 77, 14000 Valjevo, Serbia; drago@gajicassociates.com; 3Faculty of Technology Zvornik, University of East Sarajevo, Karakaj 34A, 75400 Zvornik, Republic of Srpska, Bosnia and Herzegovina; aleksandar.dosic@tfzv.ues.rs.ba

**Keywords:** extraction, carotenoids, alginate, beads, encapsulation, antioxidant activity

## Abstract

The paper was aimed at developing an ultrasound-assisted extraction of carotenoids from orange peel using olive oil as a solvent. A central composite design was used to define the optimal conditions for their extraction. Under the optimal conditions (extraction time of 35 min, extraction temperature of 42 ℃, and a liquid-to-solid ratio of 15 mL/g), the experimental and predicted values of carotenoid content were 1.85 and 1.83 mg/100 g dry weight, respectively. The agreement of these values indicated the adequacy of the proposed regression model. The extraction temperature only had a negative influence on carotenoid content. The impact of extraction parameters on the carotenoid content was decreased according to the following order: extraction time, liquid-to-solid ratio, and extraction temperature. Ca-alginate beads were prepared using the extrusion process to increase the stability and protect the antioxidant activity of olive oil enriched with carotenoids. The encapsulation efficiency and particle mean diameter were 89.5% and 0.78 mm, respectively. The presence of oil extract in Ca-alginate beads was confirmed by Fourier-transform infrared spectroscopy. The antioxidant activity of the oil enriched with carotenoids before and after encapsulation in the alginate beads was determined according to the DPPH assay.

## 1. Introduction

Citruses (oranges, grapefruit, lemons, limes, mandarins) are among the most commonly grown fruits around the world. Their production is growing every year due to increased consumer demand. The citrus processing industry produces a huge amount of waste that represents almost 50% of the weight of fresh fruit [1]. Citrus waste, including orange peel, is a source of various polyphenols, carotenoids, dietary fiber, sugar, essential oils, ascorbic acid, and significant amounts of some trace elements [2,3]. Because of this chemical composition, orange peel extracts have various pharmacological activities (antioxidant, anti-inflammatory, antibacterial, anticancer, etc.) that are important for the prevention of many human diseases [4,5]. The extraction of the aforementioned bioactive compounds from orange peel is commonly performed using conventional extraction techniques (maceration, Soxhlet extraction, hydrodistillation) [6,7]. According to the green concept, some extraction procedures that are faster, efficient, and without degradation of thermolabile compounds were developed [8,9].

The most abundant carotenoids are *α*-carotene, *β*-carotene, lutein, zeaxanthin, and *β*-cryptoxanthin in orange peel. The total carotenoid content is in the range of 11–204 mg *β*-carotene equivalents per 100 g dry weight depending on the solvent type and extraction technique [10]. Acetone [11], ionic liquids [10,12], and a mixture of hexane and isopropanol [13] are commonly used for their extraction, but they can be dangerous to the environment. Due to their non-toxicity and non-irritability, the oils are increasingly used as solvents [14]. Carotenoid intake in the organism is associated with improved immunity and reduced risk of development of degenerative diseases, such as cardiovascular diseases, cancer, and Alzheimer’s diseases [15]. The use of carotenoids in food, cosmetic, and pharmaceutical products is limited due to their hydrophobic nature and non-stability at higher temperatures, under the effect of light and oxygen, and other external factors. One of the ways to overcome the aforementioned problem is an encapsulation of carotenoids in different delivery systems [16,17,18]. The various biopolymers, such as starch [19,20], cyclodextrins [21], carrageenan [22], pectin [23], alginate [24,25,26], etc., were used as carrier materials for the encapsulation and release of carotenoids.

Alginates are anionic structural polysaccharides extracted from bacteria (*Pseudomonas* and *Azotobacter*) or brown seaweed (Phaeophyceae). They differ in total molar mass and the relative ratio of *α*-1,4-glycoside bonds between *β*-d-mannuronic (M-block) and *α*-l-guluronic (G-block) acids that make up the copolymer structure [27]. In the structure of polymers, there are blocks of alternating copolymers (GMGMGM) and homopolymers (GGGGG and MMMMM). G-blocks are twisted, whereas M-blocks have the shape of a stretched strip. Between homopolymeric M- and G-blocks, there are areas where altering M- and G-blocks occur.

Alginates dissolve in water whereby the molecules hydrate and the solution becomes viscous. The addition of divalent cations to the solution of alginates leads to the interconnection of adjacent alginate chains via ionic bridges, which causes the formation of three-dimensional gels. The most commonly used cation is Ca^2+^, since this ion enables a relatively strong interaction between polymer chains and it is not toxic. According to the properties, these gels are similar to solid materials, but they consist of 99.0–99.5% water. This biopolymer has several properties (low price, easy to use, biodegradability, and biocompatibility), which makes it a suitable material for encapsulation of extracts [28,29].

To date, no reports are available that describe the extraction of carotenoids from orange peel using oils as the solvent. In this paper, ultrasound-assisted extraction (UAE) of carotenoids from orange peel with oil was developed and optimized using a central composite design (CCD). The carotenoid content (CC) in the extract was spectrophotometrically determined (UV-Vis method). The carotenoids-enriched olive oil under optimal extraction conditions were encapsulated within Ca-alginate beads using an extrusion process. The emulsion stability, microencapsulation process yield, encapsulation efficiency, and mean particle size were investigated. The compatibility of oil enriched with carotenoids entrapped in alginate beads was evaluated through FT-IR analysis. In addition, the antioxidant activity of oil enriched with carotenoids from orange peel before and after encapsulation in the Ca-alginate beads was determined.

## 2. Materials and Methods

### 2.1. Chemicals and Reagents

In this study, cyclohexane (Sigma Chemical, St. Louis, MO, USA), alginic acid sodium salt—very low viscosity (Chem Cruz, Dallas, TX, USA), calcium chloride dehydrate (Zorka Pharma, Sabac, Serbia), and refined olive oil with additional non-refined oil (Cretan Mill EV, Nea Alikarnassos, Greece) were used.

### 2.2. Plant Material

The ground orange peel was dried at 40 ℃ for a month to a moisture content of 13.37% (*w*/*w*). The moisture was in the allowed range of 7–16% (*w*/*w*) reported in Yugoslav Pharmacopoeia 2000 (Ph. Jug. V) [30]. The drying process was performed according to the following protocol: Plant material was dried for 2 h, and then the weight was measured; after that, the drying process was carried out every 30 min up to the constant weight. Before extraction, the plant material was powdered in the electric mill and sifted through a sieve with a pore size of 0.10, 0.25, 0.40, 0.50, 0.63, 0.80, 1.00, and 1.32 mm. Plant material with a diameter of 0.50 mm was taken for additional carotenoid extraction.

### 2.3. Modeling of the UAE

The UAE of carotenoids from orange peel using olive oil was carried out in an ultrasonic bath (Sonic, Nis, Serbia) with a total volume of 6 L. The used frequency and power were 40 kHz and 150 W, respectively. The extraction time, extraction temperature, and liquid-to-solid ratio were varied at the five levels according to the matrix of CCD. The actual and coded factor levels are depicted in Table 1. In this way, the simultaneous impact of factors was considered on the CC. After the extraction, the plant material was separated by centrifugation at 4000 rpm for 15 min.

The actual factor levels were transformed to better uniformity using Equation (1):(1)$$x_{i}=\frac{X_{i}-X_{o}}{\Delta X_{i}}$$
where *x_i_* is coded values, *X_i_* is actual values, *X_o_* is the actual value in the central point, and Δ*X_i_* is the step of change.

The obtained results for CC were fitted using a second-order polynomial model. Generally, the second-order polynomial can be presented as follows (Equation (2)):(2)$$Y=a_{0}+a_{1}x_{1}+a_{2}x_{2}+a_{3}x_{3}+a_{11}x_{1}^{2}+a_{22}x_{2}^{2}+a_{33}x_{3}^{2}+a_{12}x_{1}x_{2}+a_{13}x_{1}x_{3}+a_{23}x_{2}x_{3}$$
where *Y* is the system response; *a*_0_ is the intercept; *a*_1_, *a*_2_, *a*_3_, *a*_11_, *a*_22_, *a*_33_, *a*_12_, *a*_13_, and *a*_23_ are the regression coefficients; *x*_1_, *x*_2_, and *x*_3_ are the factors, and *ε* is the residual.

### 2.4. Statistical Analysis

The data analysis was carried out using Design Expert 11.0.3.0 (Stat-Ease, Minneapolis, MN, USA). Regression analysis was used to fit the obtained data. *F*-values were calculated to estimate the adequacy of the proposed regression model.

### 2.5. Analysis of Carotenoids

CC in olive oil enriched with carotenoids was determined spectrophotometrically according to the method described by Goula et al. [14]. Exactly 3 g of oil was dissolved in 10 mL of cyclohexane and absorbance was measured at 470 nm. The CC was calculated based on Equation (3):(3)$$C=\frac{A\cdot 10^{6}}{2000\cdot 100\cdot d}$$
where *A* is the absorbance of the sample at 470 nm, *d* is the thickness of the layer through which the light beam passes, and *C* is the concentration of carotenoids in mg/kg oil.

### 2.6. Solution Preparation

The solutions of alginate (1.5%, *w*/*v*) and calcium chloride (2%, *w*/*v*) were prepared separately in distilled water. The alginate solution was then stirred and stored overnight to ensure dissolution.

### 2.7. Preparation of Alginate-Oil Emulsion

The alginate-oil emulsion was prepared according to a previously described method [31]. The olive oil enriched with carotenoids (43 mL) was gradually added to the alginate solution (100 mL) for 45 min.

### 2.8. Preparation of Oil-Loaded Ca-Alginate Beads

The alginate-oil emulsion was transferred to the plastic syringe and passed through a needle diameter of 26 G (0.45 × 12 mm). The needle was fixed at 15 cm above the surface of the gelling bath. The flow of emulsion was 33 mL/h and the pressure of secondary airflow was 0.8 bar. The formed drops of the emulsion were torn off from the needle tip thanks to the effect of gravity and secondary airflow. The gelling solution was homogenized with a magnetic stirrer (500 rpm) at room temperature (22 ± 2 ℃) to prevent the beads from sticking together. The beads were kept to harden in the gelling bath for 30 min. The empty (control) alginate microparticles were prepared in the same way, except that sodium-alginate and calcium chloride were dissolved in distilled water.

### 2.9. Characterization of Alginate-Oil Emulsion

The droplets of emulsion immediately after preparation were observed under a microscope at 10× magnification to determine the emulsion’s microstructure. Centrifugation was used to estimate the emulsion stability (ES). A total of 5 mL of the emulsion was sampled, transferred into the plastic cuvette of 10 mL, and centrifugated twice at 300 rpm for 15 min. The measurements were performed using a laboratory centrifuge (LC 320, Tehtnica, Zelezniki, Slovenia) at room temperature (20 ± 2 ℃). The emulsion stability was calculated using Equation (4):(4)$$ES\left(\%\right)=\frac{V_{e}}{V_{i}}\times 100$$
where *V_e_* is the volume of remaining emulsion after centrifugation and *V_i_* is the volume of the initial emulsion.

### 2.10. Determination of Encapsulation Yield

The encapsulation yield is a ratio of the amount of obtained beads and the amount of emulsion used in the encapsulation process. The encapsulation yield (*Y*) was calculated as follows (Equation (5)):(5)$$Y\left(\%\right)=\frac{M_{b}}{M_{em}}\times 100$$
where *M_b_* is the weight of obtained beads and *M_em_* is the weight of the used emulsion.

### 2.11. Determination of Encapsulation Efficiency

The non-encapsulated oil was determined by measuring the weight of free oil left on the surface of the gelling solution as well as the surface of the wet beads. The oil from the surface of the beads was adsorbed. Then the beds were dried in the laboratory oven to a constant weight. In Equation (6), the weight of encapsulated oil (*W*_3_) was obtained as the difference between the initial weight of oil (*W*_1_) and the weight of non-encapsulated oil before drying (*W*_2_).
(6)$$W_{3}=W_{1}-W_{2}$$

The encapsulation efficiency (EE), which represents the percentage of oil encapsulated relative to the initial amount of used oil, was calculated according to Equation (7):(7)$$EE\left(\%\right)=\frac{W_{3}}{W_{1}}\times 100$$

### 2.12. Characterization Analysis

#### 2.12.1. Shape and Size of Beads

The size and the shape of the beads were determined using an optical microscope equipped with a digital camera. The sphericity factor (SF), calculated according to Equation (8), was used to estimate the roundness of the formed beads [31]:(8)$$SF=\frac{D_{max}-D_{per}}{D_{max}+D_{per}}$$
where *D_max_* is the maximum diameter passing through the centroid of the bead (mm) and *D_per_* is the diameter perpendicular to *D_max_* passing through the centroid of the bead (mm).

The beads are a perfect sphere when the SF is around zero. A greater degree of shape distortion of the beads is achieved when the SF has higher values.

#### 2.12.2. Fourier-Transform Infrared Spectroscopy (FT-IR)

FT-IR spectra of sodium alginate powder, Ca-alginate beads, olive oil enriched with carotenoids from orange peel, and oil-loaded Ca-alginate beads were recorded after the preparation of KBr pellets. In the case of oil, a few drops of oil were positioned on the surface of KBr. The scanning of the samples was performed in the wavelength number from 4000 to 400 cm^−1^ with a resolution of 4 cm^−1^ using an FT-IR spectrophotometer (Bomem Hartmann & Braun MB-series, Quebec, QC, Canada). The data processing was carried out using Win-Bomem Easy (Galactic Industries, Salem, NH, USA).

### 2.13. Determination of Antioxidant Activity

The antioxidant activity of the oil enriched with carotenoids before and after encapsulation in Ca-alginate beads was determined using the DPPH assay [32]. The Ca-alginate beads did not significantly interfere with the determination of antioxidant activity [33]. A total of 1 g of Ca-alginate beads was dissolved in 5 mL of toluene. Instead of the oil, the equivalent amount of toluene was added to the sample of the negative control. The absorbances of the samples were measured at 517 nm relative to toluene. The antioxidant activity was estimated based on the half-maximal inhibitory concentration (IC_50_) that was presented as the mean value of three measurements ± standard deviation.

## 3. Results and Discussion

The conventional techniques require the use of organic solvent for the extraction of carotenoids from plant material. Solvent evaporation and its recycling are additional steps that should be performed after using these techniques. These facts cause the increase of high input costs, energy consumption, and hazardous waste generation of the solvent after its evaporation from the sample. The UAE gives higher yields of desired compounds for shorter extraction times and at lower temperatures compared with other conventional techniques [34]. To avoid the use of toxic and harmful solvents for the extraction of carotenoids from tomatoes and pomegranate peel, vegetable oils were used as the solvents. The oils are biodegradable and non-toxic, which slows down the oxidation time and the rate of degradation of carotenoids [35]. Additional processing operations are not required after performed extraction, and thus enriched oil can be used in food, pharmaceutical, and cosmetic products. However, the high viscosity of oils results in low diffusivity in the cells of plant material and low extraction yield even at higher temperatures. One of the ways to overcome this problem is the use of UAE of carotenoids using vegetable oils. The effect of cavitation energy leads to the destruction of plant cells and the release of intracellular substances in the solvent. Since the use of UAE and olive oil for the extraction of carotenoids from orange peel has not been described in the literature, the paper aimed to optimize this procedure using the response surface methodology. The physicochemical parameters of the olive oil used for the extraction are depicted in Table 2.

### 3.1. Optimization of the UAE of Carotenoids

The UAE of carotenoids from orange peel was optimized using CCD. This mathematical approach is more efficient compared with other optimization methods such as one-variable-at-a-time (OVAT) since optimization can be carried out based on a smaller number of experimental runs. For the optimization of three factors at five levels, it is necessary to perform 53 = 125 experimental runs according to OVAT, whereas 32 + 3 × 2 + 5 = 20 experimental runs are necessary to perform according to the CCD. In Table 3, the matrix of the CCD representing the combination of factor levels is given. The CC was varied in the range of 0.3–2.0 mg/100 g d.w. The lowest CC of 0.3 mg/100 g d.w. was obtained for 25 min at 67 °C and a liquid-to-solid ratio of 10 mL/g. The highest CC of 2.0 mg/100 g d.w. was determined in the extract prepared for 42 min at 50 °C and the liquid-to-solid ratio of 10 mL/g.

The dependence of the analyzed factors on the response (Y) can be presented as follows (Equation (9)):(9)$$Y=1.06+0.27*A-0.11*B+0.15*C-0.02*AB+0.11*AC-0.17*BC+0.17*A^{2}-0.20*B^{2}-0.05*C^{2}$$

Extraction temperature had a negative effect, whereas extraction time and liquid-to-solid ratio had a positive effect on the yield of CC. The highest effect had extraction time followed by liquid-to-solid ratio and extraction temperature.

The analysis of variance (ANOVA) of the proposed regression model at the 95% confidence level is presented in Table 4. Based on the p-values, the statistical significance of terms in the polynomial equation was estimated. The terms with a *p*-value lower than 0.05 were considered statistically significant. All terms, except the interaction between extraction time and temperature, were statistically significant in the polynomial equation. The insignificant term can be excluded from the equation since its contribution to increasing/decreasing the CC was very small. The regression model was statistically significant because its *F*-value of 45.80 was higher than the critical *F*-value of 3.02. The *F*-value of lack of fit (0.3816) was not statistically significant relative to the pure error (0.0519). These parameters indicated the adequacy of the signal in the navigated space.

The statistics of fitting for a second-order polynomial model are depicted in Table 5. The coefficient of variation was 8.42% and it should have been as low as possible. The adequate precision of 28.42 was acceptable since it was far greater than the limit value of 4. This value represents a measure of signal-to-noise ratio. The coefficient of determination (R^2^) of 0.976 indicated that 97.6% of the variation in the CC can be explained by the proposed model. The adj R^2^ of 0.955 was close to the R^2^ of 0.920, so the model can be considered precise and accurate for the prediction of the CC in the design space.

In Figure 1a, it can be noticed that Cook’s distances were lower than the limit value (1). Since the data in the plot of the normal distribution of residuals (Figure 1b) were close to the line, it can be concluded that the residuals followed a normal distribution.

The three-dimensional plots representing the effect of the extraction parameters on the CC are presented in Figure 2. The effect of extraction time was almost the same at the different levels of extraction temperatures (Figure 2a). This behavior can be explained by the fact that in the first phase of extraction, the desired compounds were leached from the surface of the damaged cell walls. The longer effect of cavitation energy damaged the cell membranes, which led to the release of carotenoids. After that period, the diffusion of carotenoids through the cell walls was also performed, but this process was much slower. Since the ultrasonic waves traveled through the medium, they caused the compression and shear of solvent molecules, resulting in localized changes in density and modulus of elasticity [36]. The initial sinusoidal waves of compression and shear were distorted into shock waves at a finite distance from the ultrasound source. A sudden decrease in pressure at the edge of the ultrasound wave created small bubbles. The ultrasound also facilitated swelling and hydration, and led to the expansion of pores in the cell wall. The diffusion through the cell walls and leaching of cellular contents were also attributed to improved extraction performance. Particle size reduction by ultrasonic disintegration increased the number of cells that were directly exposed to the extraction by the solvent and ultrasonic cavitation. The effect of extraction temperature on the CC was more pronounced at longer extraction times. The increase in temperature to 45 °C led to an increase in CC, and then this value sharply decreased. The heating of olive oil reduced the viscosity and thus enabled better penetration through the plant material and better solubility of carotenoids due to the increased diffusion coefficient [14]. Goula et al. [14] concluded that a further increase of extraction temperature negatively impacts the CC, probably as a result of dissolving the impurities and degradation of carotenoids in the plant material. In Figure 2a, the effect of extraction time is significantly pronounced at the higher liquid-to-solid ratios. A similar observation can be noticed when considering the effect of the liquid-to-solid ratio on the CC. This effect was more pronounced for longer extraction times. Based on the shape of the surface depicted in Figure 2c, it can be concluded that there was a strong interaction between the extraction temperature and liquid-to-solid ratio. The change of temperature had a significant effect at a liquid-to-solid ratio higher than 10 mL/g. The increase of liquid-to-solid ratio affected the increase in the concentration gradient during diffusion from the plant material into the solvent, which resulted in the higher yield of carotenoids.

Optimal conditions of UAE of carotenoids using olive oil from orange peel were obtained by the numerical optimization method. Using this method, the polynomial model tries to find the design space in a defined range of factors where it is possible to obtain the maximal CC. The detailed analysis showed that the extraction time of 35 min, extraction temperature of 42 °C, and the liquid-to-solid ratio of 15 mL/g were the optimal conditions. At these conditions, the regression model predicted the CC of 1.83 mg/100 g d.w., whereas the experimental value was 1.85 mg/100 g d.w. A good agreement between these values indicated the adequacy of the proposed model. After a comparison of the results obtained in this study with those available in the literature [10], it can be concluded that the choice of extraction technique and solvent (olive oil) was adequate. The CC of 2.28 mg/100 g d.w. was only higher in the case of ionic liquid 1-*n*-butyl-3-methylimidazolium tetrafluoroborate. In the case of ionic liquid 1-butyl-3-methylimidazolium chloride, the values of CC were almost the same as in this study. The other solvents, such as acetone, 1-*n*-butyl-3-methylimidazolium hexafluorophosphate, and 1-hexyl-3-methylimidazolium chloride, extracted lower CC.

### 3.2. Encapsulation Optimal Oil Extract in Ca-Alginate Beads

#### 3.2.1. Alginate-Oil Emulsion

The alginate-oil emulsion was observed under a microscope to estimate the oil droplet size (Figure 3). The oil droplet sizes were relatively similar at about 60 μm.

The determined value of ES value for the prepared alginate-oil emulsion was 84.6%. It indicated that the alginate-oil emulsion was relatively stable and the phase separation was not observed. Chan [31] confirmed that the emulsion is stable at the higher concentrations of alginate and for an oil loading up to 40 vol%. The increase in alginate concentration led to the higher viscosity of the continuous phase surrounding the oil droplets, restricting their movement. The emulsion stability has a significant influence on the encapsulation efficiency, and emulsion with a small oil droplet size could increase the retention of encapsulated oil products [29].

#### 3.2.2. Encapsulation Process Yield

The yield of encapsulation was 92.3%, which was in good agreement with a yield of 94% obtained for encapsulation of black seed oil in alginate beads [29]. The reduction in yield was probably the result of the dissipation of emulsion during the emulsification process, and due to the loss of beads during the electrospray process through the needle.

#### 3.2.3. Encapsulation Efficiency

The EE, expressed as the percentage of encapsulated oil relative to the total oil used, was 89.5%. Since the encapsulation efficiency was a fairly high value, the proposed extrusion method was adequate for the encapsulation of olive oil enriched with carotenoids from orange peel. A low amount of encapsulated oil was the result of oil loss during encapsulation. Azad et al. [29] obtained an EE of 90.1% for black seed oil in dried alginate beads.

### 3.3. Characterization Analysis

#### 3.3.1. Shape and Size of Beads

The typical size and shape of the beads are depicted in Figure 4. The size of the beads produced was about 0.78 mm in diameter.

Based on the SF, the bead sphericity was estimated. In this study, the SF of 0.03 was lower than 0.05, so the prepared beads were considered spherical. The size and sphericity of the beads commonly depends on a few factors, such as the emulsion flow rate and the distance between the needle and encapsulating medium surface [29,31]. The flow rate of 33.3 mL/h and the distance between the needle and encapsulating medium surface of 15 cm provided the production of the desired size of the spherical bead.

#### 3.3.2. FT-IR Analysis

The FT-IR analysis was used for confirmation of the established interactions between alginate and oil. FT-IR spectra of sodium alginate powder, Ca-alginate beads, oil enriched with carotenoids from orange peel, and oil-loaded Ca-alginate beads are presented in Figure 5. In the spectrum of sodium alginate (Figure 5a), the broad intensive band at 3411 cm^−1^ was the result of valence vibrations of the OH bond. The valence vibrations of the aliphatic C–H bond could be noticed at 2929 cm^−1^. The asymmetric and symmetric vibrations of the carboxylic anion (COO^−^) had bands at 1640 and 1438 cm^−1^, respectively. The valence vibrations of the C–O bond of the etheric and alcoholic groups had the bands at 1098 cm^−1^ and 1031 cm^−1^, respectively.

After cross-linking alginate with Ca^2+^ ions, the bands of stretching vibrations of O–H bonds were in a narrower range than sodium alginate (Figure 5b). This behavior was the result of the participation of the hydroxyl and carboxylate groups of alginate to the calcium ions. The formation of a chelating structure impacted the decrease in hydrogen bonds between OH groups, causing a narrower band in the spectrum of calcium alginate. The interaction between the Ca^2+^ ions and carboxyl groups of alginate resulted in a decrease in the wavelength of the COO^−^ group. The bands in the spectrum of sodium alginate at 1640 and 1438 cm^−1^ were shifted to 1636 and 1432 cm^−1^, respectively, in the spectrum of calcium alginate. The stretching vibration of the etheric group was shifted from 1031 to 1029 cm^−1^, and the decrease in the intensity of the band was also noticed in the spectrum of calcium alginate. These characteristic peaks for sodium alginate and Ca-alginate beads powder were also reported by other authors [37,38].

The spectrum of olive oil enriched with carotenoids from orange peel (Figure 5c) had characteristic bands of sp^2^ CH stretching vibration (2925 and 2854 cm^−1^), COO− stretching vibration (1747 cm^−1^), and CH_2_ bending (1466 cm^−1^). The band at 3007 cm^−1^ indicated the presence of sp^2^ C–H stretching vibration. The characteristic 7-*cis* and 15-*cis* configuration of *β*-carotene was between 1234 and 722 cm^−1^ [39]. In addition to the already confirmed bands, the band originating from the OH group of alginate was also noticed in the spectrum of oil-loaded Ca-alginate beads (Figure 5d). It indicated that there was not a strong interaction between alginate and oil encapsulated into the beads. Keeping this in mind, it can be concluded that the oil was only entrapped in the inner of the beads without chemical interactions.

### 3.4. Antioxidant Activity

The antioxidant activity of the oil enriched with carotenoids before and after encapsulation in the alginate beads was determined according to the DPPH assay. The concentration of oils before and after encapsulation was observed in the concentration range of 0.2–100 mg/mL. The IC_50_ value of oil before encapsulation was 17.04 ± 0.52 mg/mL, whereas after encapsulation it was found to be 17.01 ± 0.41 mg/mL. Based on these values, it can be concluded that the change in the IC_50_ value was negligible. This fact gives preference to the proposed procedure for the preparation of beads based on alginate, because in that way the antioxidant activity of the enriched oil is preserved.

## 4. Conclusions

Olive oil was successfully applied for UAE of carotenoids from orange peel. The extraction was modeled and optimized using CCD. The extraction time of 35 min, extraction temperature of 42 ℃, and the liquid-to-solid ratio of 15 mL/g were obtained after numerical optimization. Under these conditions, the olive oil was enriched with carotenoids and encapsulated in the Ca-alginate beads. The proposed UAE reduced consumption of energy and solvent and ensured high-quality oil. In addition, subsequent operations that referred to the separation of solvent from oil enriched with carotenoids were not necessary. The alginate-oil emulsion was stable and resulted in an EE of 89.5%. The oil-loaded beads were spherical with an average size of 0.78 mm. The results of FTIR analysis confirmed successful oil loading in Ca-alginate beads. Finally, the encapsulation process negligibly changed the antioxidant activity of oil extract. The Ca-alginate beads can be used as a natural, biodegradable, and biocompatible carrier for oil enriched with carotenoids. In this way, the stability of carotenoids can be improved without losing the antioxidant activity of the oil. These beads can be used for the preparation of advanced pharmaceutical and cosmetic products.

## Figures and Tables

**Figure 1 biomolecules-11-00225-f001:**
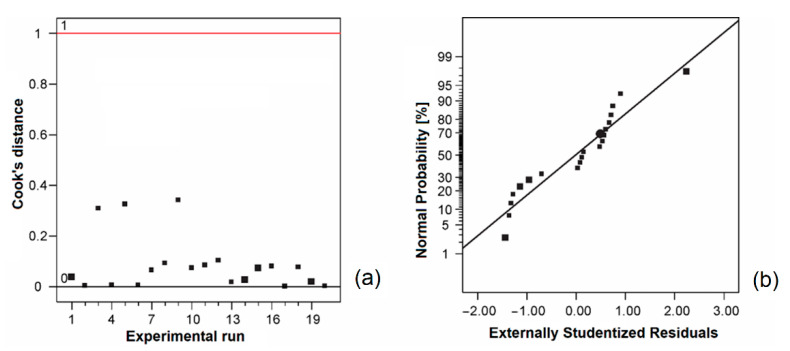
Cook’s distance (**a**) and normal distribution of residuals (**b**).

**Figure 2 biomolecules-11-00225-f002:**
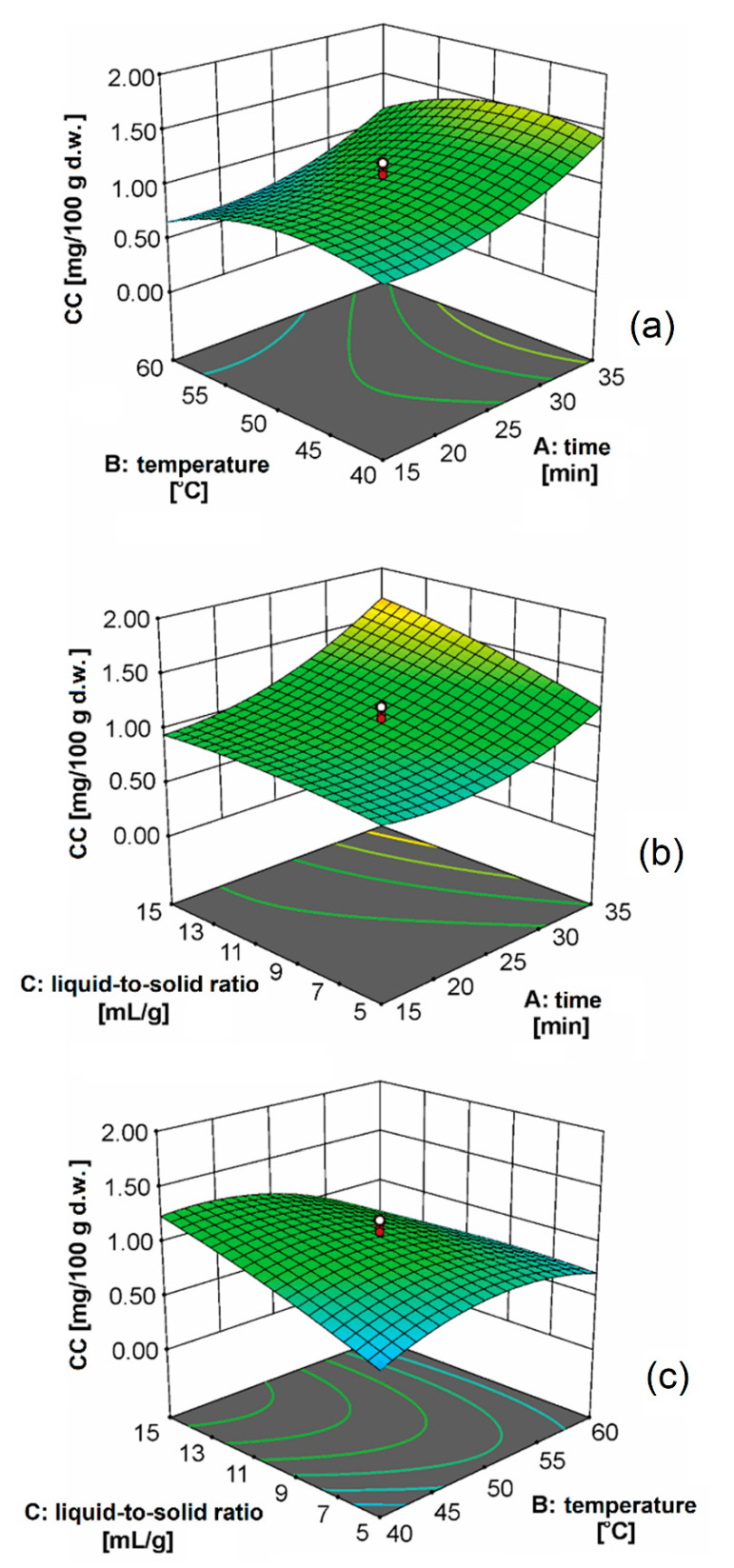
The effect of: (**a**) extraction time and temperature at the liquid-to-solid ratio of 10 mL/g, (**b**) extraction time and the liquid-to-solid ratio at 50 °C, and (**c**) extraction temperature and liquid-to-solid ratio at 20 min on the CC.

**Figure 3 biomolecules-11-00225-f003:**
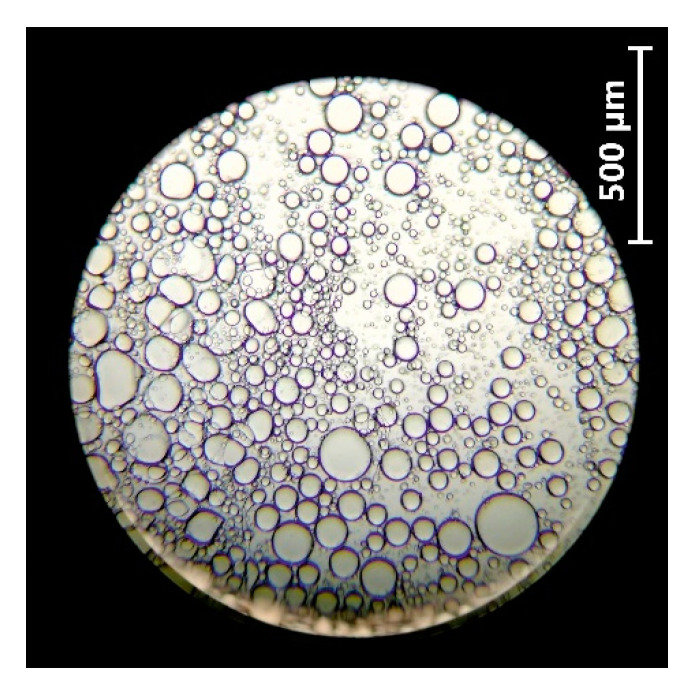
Microscopic image of freshly prepared oil droplets in the alginate-oil emulsion at a magnification of 10×.

**Figure 4 biomolecules-11-00225-f004:**
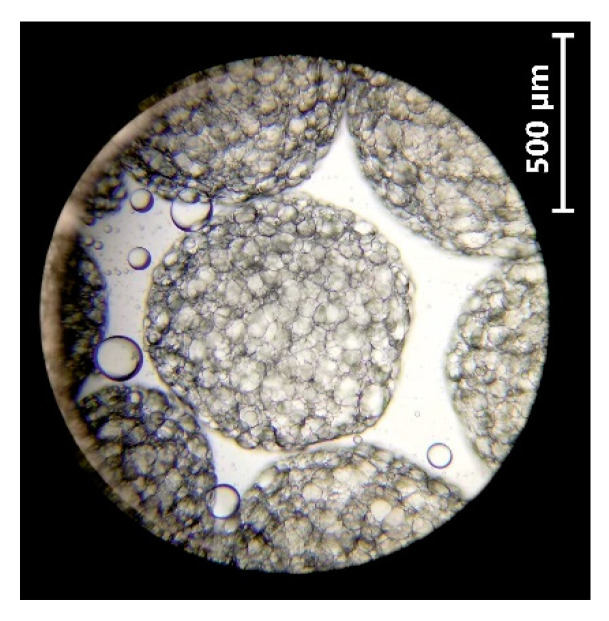
Microscopic image of oil-loaded Ca-alginate beads at a magnification of 10×.

**Figure 5 biomolecules-11-00225-f005:**
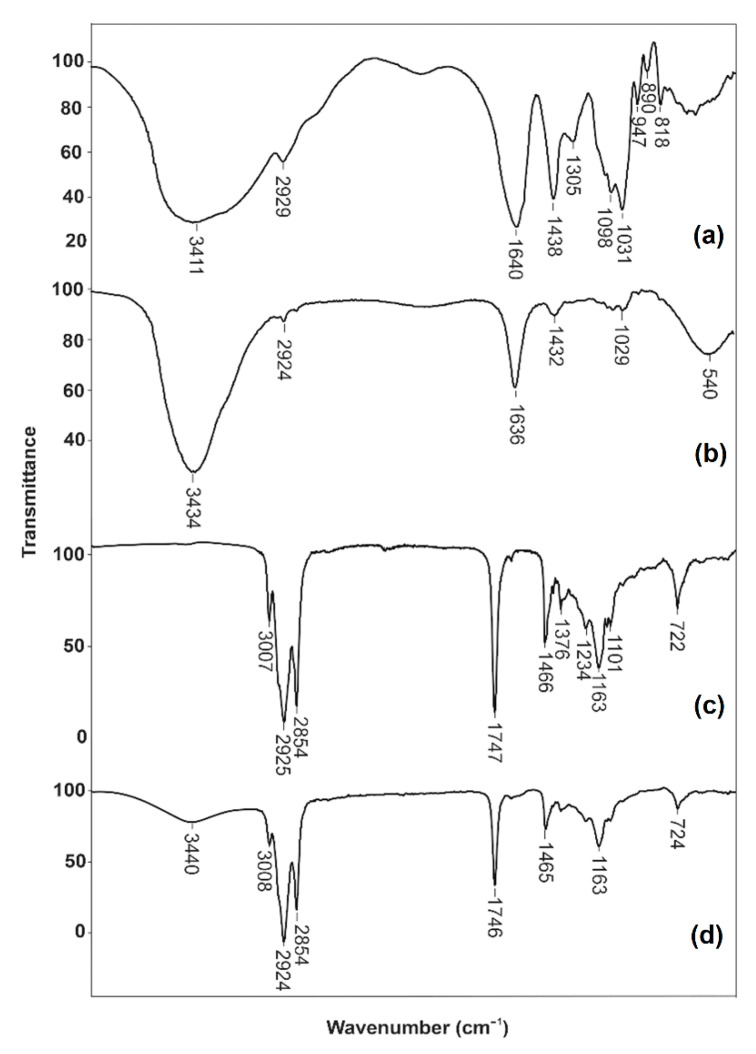
FTIR spectra of sodium alginate powder (**a**), Ca-alginate beads (**b**), oil enriched with carotenoids from orange peel (**c**), and oil-loaded Ca-alginate beads (**d**).

**Table 1 biomolecules-11-00225-t001:** The actual and coded factor levels of the used design.

Factors	Coded Values				
	−α	−1	0	+1	+α
	Actual Values				
Extraction time [min]	8	15	25	35	42
Extraction temperature [°C]	33	40	50	60	67
Liquid-to-solid ratio [mL/g]	2	5	10	15	18

**Table 2 biomolecules-11-00225-t002:** The physicochemical parameters of olive oil.

Parameters	Values
Density	0.9181 g/mL
Total fat	92.0 g/100 mL oil
Saturated fat	12.8 g/100 mL oil
Monounsaturated fat	70.5 g/100 mL oil
Polyunsaturated fat	8.3 g/100 mL oil
Vitamin E	14.0 mg/100 mL oil

**Table 3 biomolecules-11-00225-t003:** Matrix of CCD with three factors.

Std.	Run	Factor 1	Factor 2	Factor 3	Response
		A: Time [min]	B: Temperature [°C]	C: Solvomodul [mL/g]	CC [mg/100 g d.w.]
17	1 *	25 (0)	50 (0)	10 (0)	0.96
12	2	25 (0)	67 (+1.68)	10 (0)	0.30
6	3	35 (+1)	40 (−1)	15 (+1)	1.75
20	4 *	25 (0)	50 (0)	10 (0)	1.10
7	5	15 (−1)	60 (+1)	15 (+1)	0.42
13	6	25 (0)	50 (0)	2 (−1.68)	0.69
3	7	15 (−1)	60 (+1)	5 (−1)	0.78
9	8	8 (−1.68)	50 (0)	10 (0)	1.13
1	9	15 (−1)	40 (−1)	5 (−1)	0.52
2	10	35 (+1)	40 (−1)	5 (−1)	0.99
11	11	25 (0)	33 (−1.68)	10 (0)	0.72
4	12	35 (+1)	60 (+1)	5 (−1)	1.00
16	13 *	25 (0)	50 (0)	10 (0)	1.13
15	14 *	25 (0)	50 (0)	10 (0)	0.97
19	15 *	25 (0)	50 (0)	10 (0)	1.21
8	16	35 (+1)	60 (+1)	15 (+1)	1.23
5	17	15 (−1)	40 (−1)	15 (+1)	1.02
14	18	25 (0)	50 (0)	18 (+1.68)	1.22
18	19 *	25 (0)	50 (0)	10 (0)	0.99
10	20	42 (+1.68)	50 (0)	10 (0)	2.00

***** Central point of the design; CC—carotenoid content; Std.—standard order of the experiments.

**Table 4 biomolecules-11-00225-t004:** ANOVA for a second-order polynomial model.

Parameter	Sum of Squares	df	Mean	*F*-Value	*p*-Value
Model	2.9553	9	0.3284	45.80	5.94 × 10^−7^
A	1.0005	1	1.0005	139.54	3.39 × 10^−7^
B	0.1787	1	0.1787	24.92	5.4 × 10^−4^
C	0.2967	1	0.2967	41.38	7.51 × 10^−5^
AB	0.0039	1	0.0039	0.54	0.4792
AC	0.0900	1	0.0900	12.56	5.32 × 10^−3^
BC	0.2391	1	0.2391	33.35	1.8 × 10^−4^
A^2^	0.4134	1	0.4134	57.66	1.85 × 10^−5^
B^2^	0.5961	1	0.5961	83.14	3.68 × 10^−6^
C^2^	0.0306	1	0.0306	4.27	6.6 × 10^−2^
Residual	0.0717	10	0.0072		
Lack of fit	0.0198	5	0.0040	0.38	0.8431
Pure error	0.0519	5	0.0104		
Total correction	3.0270	19			

A—extraction time; B—extraction temperature; C—liquid-to-solid ratio; df—degree of freedom.

**Table 5 biomolecules-11-00225-t005:** Statistics of fitting for a second-order polynomial model.

Std. Dev.	0.08	R^2^	0.976
Mean	1.01	adj R^2^	0.955
C.V. [%]	8.42	pred R^2^	0.920
		Adequate precision	28.42

Std. dev.—standard deviation; C.V.—coefficient of variation; R^2^—coefficient of determination.

## Data Availability

The data presented in this study are available on request from the corresponding author.

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
