# Peer review of "Ultrasound-Assisted Extraction of Carotenoids from Orange Peel Using Olive Oil and Its Encapsulation in Ca-Alginate Beads"

_biomolecules, 2021, doi:10.3390/biom11020225_

Round 1

Reviewer 1 Report

This paper presents Ultrasound-Assisted Extraction of Carotenoids from Orange 2
Peel using Olive Oil and its Encapsulation in Ca-Alginate 3 Beads. I have found the results very interesting for readers and for this journal. 

only one questions: how did you measure DPPH with microbeads in solution? with any interferences?

After a fine revision of English i think that it can be published on biomolecules. 

Author Response

How did you measure DPPH with microbeads in solution? With any interferences?

RESPONSE: Thank you for this remark. We inserted the following sentence “Ca-alginate beads have not significantly interfered with the determination of antioxidant activity.” and provided the relevant reference that refers to this fact.

Reviewer 2 Report

"Abstract: The paper was aimed to develop..." - An abstract begins with a little introduction and only then the aim(s) of the work, the methods, results and conclusions. Please, fix the abstract.

Line 24 - "before/after..." - It was before or after, or both? It is important because the abstract should give a brief and precise notion about the work that will be described.

Line 55 - "One of them are natural polysaccharides, such as starch [19,20], cyclodextrins [21], carrageenan [22], pectin [23], alginate [24-26], etc. can be found." - Please, improve the language.

Line 87-90 - Please, finish the sentence.

Line 309 - "Figure 3b." - The figure is only 3, not 3b. Please, fix this. A bar of measure in the image would be good.

Line 330 - "A similar EE was obtained for the encapsulation of black seed oil..." - How much is the measure "similar EE" mentioned?

Line 335 - "The size of the beads produced was about 0.78 mm in diameter." - The bar of measure in the figure is needed.

"Acknowledgments: In this section, you can acknowledge any support given which is not covered by the author contribution or funding sections. This may include administrative and technical support, or donations in kind (e.g., materials used for experiments)." - Please, correct this topic because this is the standard text used in the template.

Author Response

Line 24 - "before/after..." - It was before or after, or both? It is important because the abstract should give a brief and precise notion about the work that will be described.

RESPONSE: We determined the antioxidant activity of oil before and after encapsulation into the calcium beads. Thank you for this remark. We corrected that in the manuscript.

Line 55 - "One of them are natural polysaccharides, such as starch [19,20], cyclodextrins [21], carrageenan [22], pectin [23], alginate [24-26], etc. can be found." - Please, improve the language.

RESPONSE: We improved this sentence in the manuscript.

Line 87-90 - Please, finish the sentence.

RESPONSE: We finished the sentence. Thank you for this remark.

Line 309 - "Figure 3b." - The figure is only 3, not 3b. Please, fix this. A bar of measure in the image would be good.

RESPONSE: Thank you for the given comment. We deleted this sentence because it referred to Figure 4.

Line 330 - "A similar EE was obtained for the encapsulation of black seed oil..." - How much is the measure "similar EE" mentioned?

RESPONSE: We provided literature data of the EE and retyped this sentence.

Line 335 - "The size of the beads produced was about 0.78 mm in diameter." - The bar of measure in the figure is needed.

RESPONSE: The bars of measure are now depicted in Figure 3 and Figure 4.

"Acknowledgments: In this section, you can acknowledge any support given which is not covered by the author contribution or funding sections. This may include administrative and technical support, or donations in kind (e.g., materials used for experiments)." - Please, correct this topic because this is the standard text used in the template.

RESPONSE: We deleted this part because it is only taken from the manuscript template.

Reviewer 3 Report

The Authors present an ultrasound-based extraction method of orange peel carotenoids using olive oil as green solvent, followed by encapsulation using calcium alginate beads.

This study is in the current trends in green chemistry.

The paper is well written, and the presentation of the results clear.

I have some minor comments to be considered before this work can be accepted:

  1. Table 3 provide SD for response and add statistical analysis.
  2. Figure 3 and 4: provide a scale bar.
  3. IC50, provide SD for each value.

Author Response

Table 3 provide SD for response and add statistical analysis.

IC50, provide SD for each value.

RESPONSE: The central composite design is a mathematical tool that already has the repeating the central point of the design so that the experimental data ± standard deviation is not necessary to perform for all experimental runs. In the case of IC50, we provided the standard deviation for both values as you suggested.

Figure 3 and 4: provide a scale bar.

RESPONSE: The bars of measure are now depicted in Figure 3 and Figure 4.